

# Inhibitory effects of $\alpha$-Mangostin on T cell cytokine secretion *via* ORAI1 calcium channel and K$^+$ channels inhibition

Hyun Jong Kim[1,2,*], Seorin Park[1,*], Hui Young Shin[1], Yu Ran Nam[2], Phan Thi Lam Hong[1,2], Young-Won Chin[3], Joo Hyun Nam[1,2] and Woo Kyung Kim[2,4]

[1] Department of Physiology, Dongguk University College of Medicine, Gyeong-ju, Gyeongsangbuk-do, Republic of Korea
[2] Channelopathy Research Center (CRC), College of Medicine, Dongguk University, Goyang, Gyeonggi-do, Republic of Korea
[3] College of Pharmacy and Research Institute of Pharmaceutical Sciences, Seoul National University, Seoul, Republic of Korea
[4] Department of Internal Medicine Graduate School of Medicine, Dongguk University, Goyang, Gyeonggi-do, Republic of Korea
[*] These authors contributed equally to this work.

Corresponding authors
Joo Hyun Nam,
jhnam@dongguk.ac.kr
Woo Kyung Kim,
wk2kim@naver.com

## ABSTRACT

**Background**. As one of the main components of mangosteen (*Garcinia mangostana*), a tropical fruit, $\alpha$-mangostin has been reported to have numerous pharmacological benefits such as anti-cancer, anti-inflammatory, and anti-allergic effects through various mechanisms of action. The effects of $\alpha$-mangostin on intracellular signaling proteins is well studied, but the effects of $\alpha$-mangostin on ion channels and its physiological effects in immune cells are unknown. Generation of intracellular calcium signaling is a fundamental step for T cell receptor stimulation. This signaling is mediated not only by the ORAI1 calcium channel, but also by potassium ion channels, which provide the electrical driving forces for generating sufficient calcium ion influx. This study investigated whether $\alpha$-mangosteen suppress T cell stimulation by inhibiting ORAI1 and two kinds of potassium channels (K$_v$1.3 and K$_{Ca}$3.1), which are normally expressed in human T cells.

**Methods**. This study analyzed the inhibitory effect of $\alpha$-mangostin on immune cell activity via inhibition of calcium and potassium ion channels expressed in immune cells.

**Results**. $\alpha$-mangostin inhibited ORAI1 in a concentration-dependent manner, and the IC$_{50}$ value was $1.27 \pm 1.144$ $\mu$M. K$_v$1.3 was suppressed by $41.38 \pm 6.191\%$ at 3 $\mu$M, and K$_{Ca}$3.1 was suppressed by $51.16 \pm 5.385\%$ at 3 $\mu$M. To measure the inhibition of cytokine secretion by immune cells, Jurkat T cells were stimulated to induce IL-2 secretion, and $\alpha$-mangostin was found to inhibit it. This study demonstrated the anti-inflammatory effect of $\alpha$-mangostin, the main component of mangosteen, through the regulation of calcium signals.

## INTRODUCTION

Calcium acts as a secondary messenger in most cells, and is important for immune responses such as immune cell activation and differentiation, cytokine production, and phagocytosis (*Oh-hora & Rao, 2008*). When the T cell receptor (TCR) is stimulated, phosphatidyl inositol 4,5-biphosphate (PIP$_2$) is hydrolyzed into inositol 1,4,5-triphosphate (IP$_3$) and diacylglycerol (DAG). IP$_3$ binds to the IP$_3$ receptor (IP$_3$R) present in the endoplasmic reticulum (ER), resulting in calcium depletion in the ER store (*Prakriya & Lewis, 2015*; *Parekh & Putney, 2005*). When the calcium in the ER store is depleted, the ORAI1 ion channels open resulting in calcium inflow (*Feske et al., 2006*). Stromal interaction molecule (STIM) was known to act as a calcium sensor in the ER calcium reservoir in 2005, and STIM, which recognizes ER calcium depletion, form puncta and binds directly to ORAI to regulate the opening and closing of channels (*Prakriya & Lewis, 2015*; *Roos et al., 2005*; *Liou et al., 2005*). ORAI and STIM are highly expressed in lymphocytes or mast cells, and play a crucial role in the immune response via Ca$^{2+}$ influx. In fact, patients with mutations or deficiencies in ORAI or STIM are known to develop severe immunodeficiency (*Feske et al., 2006*; *Feske, Wulff & Skolnik, 2015*; *Feske, 2007*). Another ion channel, which regulates the calcium signal besides ORAI in immune cells, is the K$^+$ channel. The K$^+$ channels expressed in most immune cells and regulating cell membrane potential are a voltage-gated potassium channel shaker-related subfamily, member 1.3 (also known as KCNA3 or K$_V$1.3) and the potassium intermediate/small conductance calcium-activated channel, subfamily N, member 4 (also known as KCNN4 or K$_{Ca}$3.1). They regulate hyperpolarizing the depolarized cell membrane through calcium influx (*Feske, Wulff & Skolnik, 2015*; *Cahalan & Chandy, 2009*). These are important in maintaining or enhancing calcium influx via ORAI in immune cells. When ORAI1 is activated and the cell membrane voltage is depolarized due to Ca$^{2+}$ influx, this influx is limited as the driving force is weakened by the electrochemical gradient. At this time, K$_{Ca}$3.1 is activated by the depolarized cell membrane voltage. Moreover, K$_{Ca}$3.1 is activated by increased calcium in the cell, and the cell membrane voltage is repolarized to maintain continuous calcium influx for immune cell activation (*Feske, Wulff & Skolnik, 2015*; *Cahalan & Chandy, 2009*; *Panyi, 2005*).

Mangosteen (*Garcinia mangostana*) is a tropical fruit grown in Southeast Asia. Mangosteen juice is used as a folk remedy to relieve dehydration, dysentery, and diarrhea (*Nabandith et al., 2004*). The skin of mangosteen contains several xanthone series. Thus far, 68 xanthone-type compounds have been found in mangosteen, including $\alpha$, $\beta$, and $\gamma$-mangostin (*Chin & Kinghorn, 2008*). Among them, $\alpha$-mangostin is known to be the most important component. Since $\alpha$-mangostin was first identified by Schmid in 1855, various pharmacological effects such as anticancer, antiviral, and antioxidant activity have been proven by several researchers worldwide over the past decades (*Shan et al., 2011*; *Aizat et al., 2019*; *Ovalle-Magallanes, Eugenio-Pérez & Pedraza-Chaverri, 2017*). Several reports have also demonstrated the anti-inflammatory effect of $\alpha$-mangostin (*Chen, Yang & Wang, 2008*; *Mohan et al., 2018*; *Gutierrez-Orozco et al., 2013*; *Chae et al., 2012*).

As described above, intracellular calcium signals are important for immune cell activity; however, there are no studies related to ion channels, except for that on $\alpha$-mangostin conducted by *Itoh et al. (2008)*. Therefore, this study aimed to confirm the pharmacological effect of $\alpha$-mangostin on $Ca^{2+}$ and $K^+$ channels associated with calcium signaling in immune cells.

## MATERIALS & METHODS

### Cell culture

Human embryonic kidney 293 T (HEK293T) and Jurkat T cells were purchased from the American Type Culture Collection (Manassas, VA, USA). HEK293T cells were cultured in a 10% $CO_2$ incubator at 37 °C in Dulbecco's modified Eagle's medium (DMEM, Welgene, Gyeongsan, Korea). The culture medium contained 10% fetal bovine serum (FBS, Welgene) and 1% penicillin/streptomycin (P/S, Hyclone). Jurkat T cells were cultured in a 5% $CO_2$ incubator at 37 °C, in RPMI1640 medium (Gibco, Thermo Fisher Scientific) supplemented with 10% FBS and 1% P/S.

### Transient transfection

To measure the ORAI1 current, HEK293T cells were co-transfected with the human ORAI1 (hORAI1) and human STIM1 (hSTIM1) vector, which were purchased from Origene Technologies (Rockville, MD, USA). Transfection was performed using Turbofect (Thermo Scientific, Waltham, MA, USA) according to the manufacturer's protocol, and green fluorescence protein (pEGFP-N1, Life Technologies) was injected at a 10:1 ratio for labeling of transfected cells.

### Cell cytotoxicity

Cell viability was determined using the Cell Counting Kit 8 (CCK-8). Sample preparation and analysis were performed according to the manufacturer's protocol. Jurkat T cells were prepared and $2 \times 10^4$ cells/well were seeded in a 96-well microtiter plate, treated with $\alpha$-mangostin 0.01 $\mu$M to 10 $\mu$M and incubated for 72 h. Following treatment, 10 $\mu$L of CCK-8 per 100 $\mu$L of culture medium was added into each well, incubated for 3 h at 37 °C, and the absorbance measured at 450 nm.

### Electrophysiology

ORAI1 was measured using transiently transfected HEK293T cells, and the $K_V 1.3$ current in Jurkat T cells was measured directly. A stable cell line, in which the corresponding ion channels were overexpressed, was used for $K_{Ca}3.1$. Recording and analysis of the whole-cell patch clamp for $I_{ORAI1}$ has been previously reported (*Kim et al., 2018*). The composition of the extracellular fluid for recording $K_V 1.3$ current ($I_{KV}$) and $K_{Ca}3.1$ current ($I_{KCa}$) was 145 mM NaCl, 3.6 mM KCl, 10 mM 4-(2-hydroxyethyl)-1-piperazineethanesulfonic acid (HEPES), 5 mM glucose, 1.3 mM $CaCl_2$, and 1 mM $MgCl_2$; pH 7.4 adjusted with NaOH. The composition of the internal solution was 5 mM NaCl, 140 mM KCl, 10 mM HEPES, 5 mM ethylene glycol-bis($\beta$-aminoethyl ether)-N,N, N′, N′-tetraacetic acid, 2 mM Mg-ATP, 4.37 mM $CaCl_2$, and 0.5 mM $MgCl_2$; pH 7.2, adjusted with KOH.

The intracellular calcium concentration for recording $I_{KCa}$ was titrated to 1 µM and calculated using WEBMAXC (Stanford University, https://somapp.ucdmc.ucdavis.edu/pharmacology/bers/maxchelator/webmaxc/webmaxcS.htm). All chemicals were purchased from Sigma-Aldrich (St. Louis, MO, USA). Stock solutions and were prepared in dimethyl sulfoxide (DMSO). All stock solutions were stored at −20 °C.

## Cytokine assay

Jurkat T cells were stimulated with anti-CD3 (Peprotech, Rocky Hill, NJ) and anti-CD28 (Peprotech) to induce the secretion of interleukin-2 (IL-2). Briefly, 50 µL/well of anti-CD3 at a concentration of 5 µg/mL was added to a 96-well plate, incubated at 37 °C for 3 h, and washed three times with Dulbecco's phosphate-buffered saline (DPBS). Jurkat T cells were seeded at a density of $5 \times 10^5$ cells/well. Thereafter 2 µg/mL anti-CD28 was added into each well and cultured in a 5% $CO_2$ incubator at 37 °C for 72 h. The culture solution was subsequently collected and diluted 1:3 with DMEM. The total amount of IL-2 secreted by Jurkat T cells was measured using the IL-2 ELISA kit (Peprotech) according to the manufacturer's protocol.

## Fura-2 Ca²⁺ imaging

$[Ca^{2+}]_i$ was measured using the fluorescent $Ca^{2+}$ indicator, fura-2 acetoxymethyl ester (Fura-2 AM; Thermo Fisher Scientific). Jurkat T cells were incubated with normal Tyrode (NT) solution (145 mM NaCl, 10 mM HEPES, 5 mM glucose, 3.6 mM KCl, 2 mM CaCl₂, and 1 mM MgCl₂ ; pH 7.4, adjusted with NaOH), containing 2 µM Fura-2 AM for 30 min at 37 °C; and subsequently washed twice with NT. The cells were attached to a perfusion chamber with a 0.17 mm-thick glass bottom on an inverted microscope (Nikon eclipse Ti, Nikon, Osaka, Japan). Calcium signals were induced in the attached cells and the fluorescence was measured using an illuminator (pE-340 fura, CoolLED, Andover, UK) and recorded using a camera (sCMOS pco.edge 4.2, PCO, Kelheim, Germany). The cells were exposed to a flow excitation wavelength of 380 nm for 30 ms and an excitation wavelength of 340 nm for 100 ms; the emission was recorded at a wavelength of 510 nm. Images were recorded and were analyzed using NIS-Element AR Version 5.00.00 (Nikon).

## Statistical analysis

Data analysis was carried out using GraphPad prism 6.0 (GraphPad) and Origin 8.0 (Microcal). Data were expressed as the mean ± standard error of the mean (SEM). Bonferroni multiple comparison analysis was used following a one-way analysis of variance (ANOVA) multiple concentrations of components and IL-2 analysis. A $p$-value was $<0.05$ was considered statistically significant.

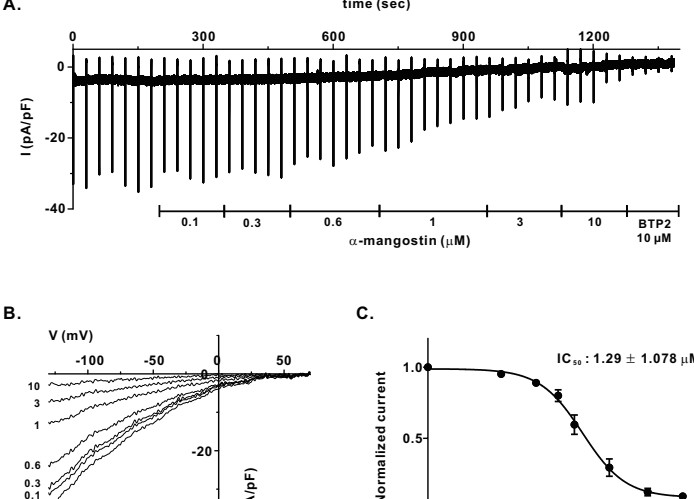

**Figure 1 Inhibitory effects of $\alpha$-mangostin on ORAI1 current ($I_{ORAI}$) in HEK293T cells co-expressed with ORAI1 and STIM1.** (A) Representative chart trace documenting $I_{ORAI1}$ inhibition of $\alpha$-mangostin. Inhibitory effects of 0.1–10 $\mu$M $\alpha$-mangostin and BTP2. (B) The current (I)-voltage (V) relationship curve of $I_{ORAI1}$ suppressed by varying concentrations of $\alpha$-mangostin. (1) control, (2) $\alpha$-mangostin at 0.1 $\mu$M, (3) 0.3 $\mu$M, (4) 0.6 $\mu$M, (5) 1 $\mu$M, (6) 3 $\mu$M, (7) 10 $\mu$M ($n$ £ 8). (C) Concentration-dependent $I_{ORAI1}$ inhibition by $\alpha$-mangostin at −120 mV, and fitted dose-response curves. Data are expressed as the mean ± SEM.

## RESULTS

### Inhibitory effect of $\alpha$-mangostin on ORAI1

We measured $I_{ORAI1}$ using the whole-cell patch clamp technique following hORAI1 overexpression in HEK293T cells by transient transfection to determine whether $\alpha$-mangostin inhibits $I_{ORAI1}$. Calcium stored in the ER to induce $I_{ORAI1}$ activity was depleted in IP$_3$ contained in the pipette solution. When the induced current was stable, it was treated with $\alpha$-mangostin to confirm its inhibitory effect. $\alpha$-Mangostin inhibited $I_{ORAI1}$ in a concentration-dependent manner (Figs. 1A–1B). Figure 1A shows the chart trace of $I_{ORAI1}$, and Fig. 1B shows the inhibition by $\alpha$-mangostin as a current–voltage relationship curve. Figure 1C shows the $I_{ORAI1}$ half-maximal inhibitory concentrations (IC$_{50}$) of $\alpha$-mangostin, with an IC$_{50}$ of 1.27 ± 1.144 $\mu$M. To confirm that ORAI1 inhibits intracellular calcium signaling, we measured the intracellular Ca$^{2+}$ concentration ([Ca$^{2+}$]$_i$) in Jurkat T cells using the fluorescent dye Fura-2. To activate ORAI1, the ER was depleted using thapsigargin, an SERCA pump inhibitor. Upon changing from 0 Ca$^{2+}$ to 2 mM Ca$^{2+}$ following thapsigargin treatment, calcium influx by ORAI1 occurs. When calcium influx was maintained at a constant level, it was treated with 1 $\mu$M and 3 $\mu$M $\alpha$-mangostin (Fig. 2A); [Ca$^{2+}$]$_i$ was inhibited by $\alpha$-mangostin by 23.90 ± 12.501% and 77.14 ± 6.600%, respectively (Fig. 2B).

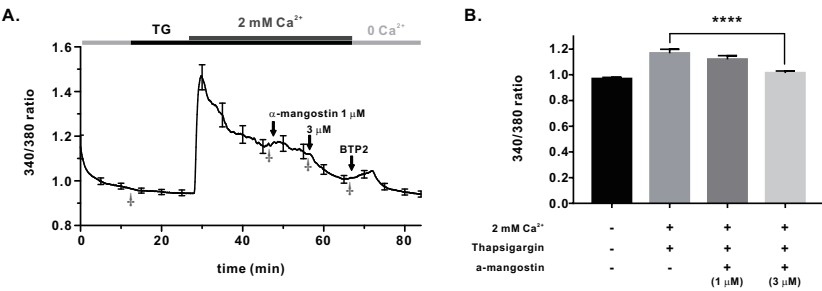

**Figure 2** Store operated Ca²⁺ entry (SOCE) induced by thapsigargin in Jurkat T cells and the inhibitory effects of *α*-Mangostin on SOCE. SOCE was induced with thapsigargin, and the inhibitory effects were confirmed by treatment with 1 μM and 3 mM *α*-mangostin. BTP2 was used as a positive control. (A) Average trace (*n* = 18) shows changes in the intracellular calcium signal by *α*-mangostin in Jurkat T cells stimulated with thapsigargin. (B) Average value at the stabilization point of the calcium signal. The average value of points marked with †. Data are expressed as the mean ± SEM. **** *p* < 0.0001.

## Inhibitory effect of *α*-mangostin on $K_V1.3$ and $K_{Ca}3.1$, the regulators of calcium signaling in immune cells

We measured the activity of potassium channels, which is necessary for maintaining the membrane driving force for sufficient calcium influx via ORAI1, and examined whether *α*-mangostin could inhibit it. It has been reported that $K_V1.3$ expression increases when the T-cell receptors (TCRs) of T-lymphocytes are stimulated (*Decoursey et al., 1987*). Therefore, Jurkat T cells were treated with 3 μg/mL anti-CD3 (Peprotech) for 24 h, and $I_{KV}$ subsequently measured using the whole-cell patch clamp technique. When the cell membrane voltage was changed from -120 mV to +60 mV for 500 ms, $I_{KV}$ increased from -60 mV (Fig. 3A). It was confirmed that $I_{KV}$ decreased by 41.38 ± 6.191% at 3 μM (Fig. 3B) upon *α*-mangostin treatment when the magnitude of the current remained stable. Finally, it was confirmed that the current was completely reduced by treatment with PAP-1, an inhibitor of $K_V1.3$ (Fig. 3B). In addition, we examined the effect of *α*-mangostin on $K_{Ca}3.1$, which is activated when calcium signals are generated in T cells (*Lam & Wulff, 2011*). However, in Jurkat T cells, $K_{Ca}3.1$ expression was extremely low and the measured current was extremely small, which is not suitable for analyzing the inhibitory effect of *α*-mangostin. Therefore, whole-cell patch clamp was performed using the stable cell line overexpressing $K_{Ca}3.1$. As $I_{KCa}$ is activated by an increase in intracellular calcium concentration, the intracellular calcium concentration was fixed at 1 μM. When $I_{KCa}$, activated by a fixed calcium concentration, was maintained at a constant level, it was inhibited by *α*-mangostin treatment in a concentration-dependent manner. As shown in Fig. 4A, *α*-mangostin inhibited $I_{KCa}$ by 28.28 ± 5.412% and 51.16 ± 5.385% at concentrations of 1 μM and 3 μM, respectively. Normalized data are summarized in Fig. 4B.

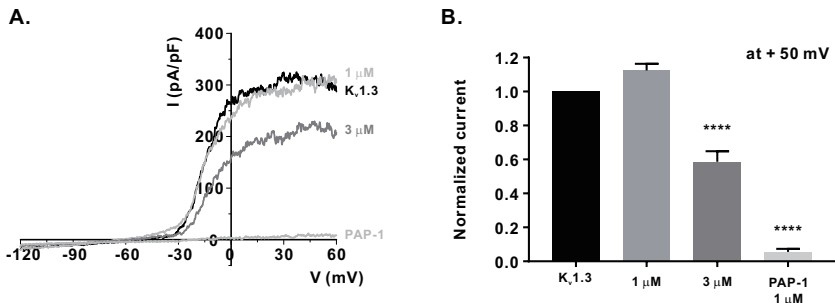

**Figure 3** Inhibitory effects of $\alpha$-mangostin on $K_V1.3$ current ($I_{KV}$) measured in Jurkat T cells. (A) The representative current (I)-voltage (V) relationship curve showing the inhibition of $I_{KV}$ by $\alpha$-mangostin ($n \pounds 7$). (B) Average value of current measured at +50 mV. Current without $\alpha$-mangostin treatment and reduced current with $\alpha$-mangostin treatment were compared. Data are expressed as the mean $\pm$ SEM. **** $p < 0.0001$.

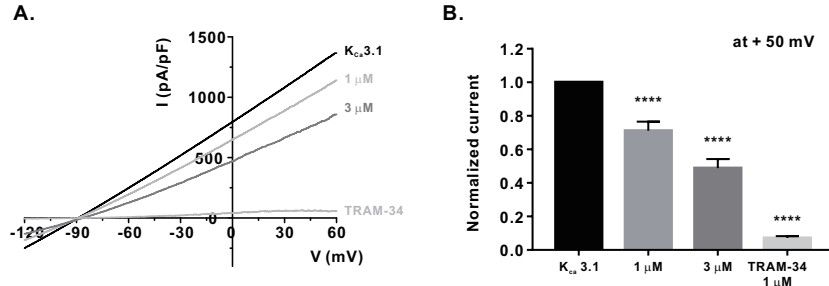

**Figure 4** Inhibitory effects of $\alpha$-mangostin on $K_{Ca}3.1$ current ($I_{KCa}$) measured in HEK293T cells over-expressed with $K_{Ca}3.1$. (A) The representative current (I)-voltage (V) relationship curve showing the inhibition of $I_{KCa}$ by $\alpha$-mangostin ($n \pounds 7$). (B) Average value of current measured at +50 mV. Current without $\alpha$-mangostin treatment and reduced current with $\alpha$-mangostin treatment were compared. Data are expressed as the mean $\pm$ SEM. **** $p < 0.0001$.

## Inhibitory effect of $\alpha$-mangostin on IL-2 production in Jurkat T cells stimulated by anti-CD3/anti-CD28

Finally, we investigated whether $\alpha$-mangostin inhibits cytokine secretion. First, the cytotoxicity of $\alpha$-mangostin was evaluated. Following $\alpha$-mangostin treatment of Jurkat T lymphocytes, cell viability was measured. Most cells died, when treated with $\alpha$-mangostin at a concentration of 10 $\mu$M, however, >80% survived when treated with $\alpha$-mangostin at a concentration of 3 $\mu$M (Fig. 5A). Thus, we subsequently analyzed whether the cytokine secretion of Jurkat T cells was inhibited at a maximum concentration of 3 $\mu$M or less. Consequently, $\alpha$-mangostin inhibited IL-2 production in a concentration-dependent manner, and the highest inhibition rate ($80.14 \pm 3.987\%$) was observed at 3 $\mu$M (Fig. 5B).

## DISCUSSION

In this study, we investigated whether $\alpha$-mangostin inhibits the calcium ion channels that cause intracellular calcium increase in immune cells and the potassium channels

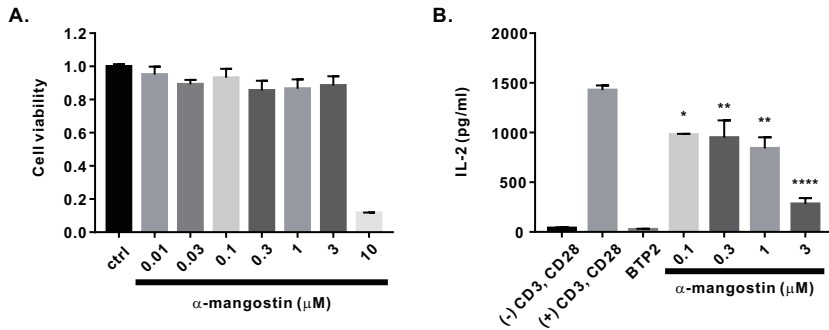

**Figure 5** **Inhibitory effects of $\alpha$-mangostin on IL-2 secretion in Jurkat T cells co-stimulated by CD3 and CD28.** (A) Cell viability analyzed after 72 h of treatment with $\alpha$-mangostin in Jurkat T cells. An equal volume of DMSO was used to compare the effect of the solvent ($n = 3$). (B) Concentration-dependent inhibitory effects of $\alpha$-mangostin in Jurkat T cells co-stimulated with anti-CD3/anti-CD28. BTP2 was used as a positive control. Data are expressed as the mean $\pm$ SEM. *$p < 0.05$, **$p < 0.01$, **** $p < 0.0001$.

that play a crucial role in regulating the cell membrane voltage, and whether cytokine production can be suppressed by it. Several studies have reported the anti-inflammatory effects of $\alpha$-mangostin. Inducible nitrogen oxide synthase (iNOS) is an enzyme that causes inflammation, and $\alpha$-mangostin and $\gamma$-mangostin have been reported to inhibit the production of NO and PEG2 in LPS-induced RAW264.7 cells through inhibition of iNOS expression (*Chen, Yang & Wang, 2008*; *Mohan et al., 2018*). In 2013, Falbiola reported that $\alpha$-mangostin inhibited the secretion of inflammatory mediators in various human cell lines, thereby exhibiting anti-inflammatory effects (*Gutierrez-Orozco et al., 2013*). Immune cells generate calcium signals due to antigen stimulation and trigger various immune responses. In 2012, Chin reported that $\alpha$-mangostin inhibited degranulation induced by A23187 and PMA in bone marrow mast cells, inhibited the production of IL-6, prostaglandin D$_2$ (PGD$_2$), and leukotriene, and reduced the expression of COX-2 mRNA, thereby having anti-allergic effects (*Chae et al., 2012*). As such, the anti-inflammatory benefits of $\alpha$-mangostin by various mechanisms have been reported; however, only a few studies have focused on the ion channels related to the generation of calcium signals.

## Effects of $\alpha$-mangostin on the calcium channel (ORAI1)

TCR stimulation by antigens activates T cells and triggers calcium signaling. This increased calcium is known to be involved in T cell proliferation and the production and secretion of cytokines (*Feske, Wulff & Skolnik, 2015*; *Putney , 2012*). TCR stimulation activates ORAI1, and the intracellular calcium increase by this process binds to Ca$^{2+}$-modulated protein (calmodulin) to generate various downstream signals. Calcium-calmodulin activates the nuclear factor of activated T cells (NFAT), thereby regulating cytokine production and immune cell proliferation (*Trebak & Kinet, 2019*; *Vaeth & Feske, 2018*). Ion channels that generate calcium signals in immune cells include ORAI, transient receptor potential (TRP) channels, and voltage-dependent Ca$^{2+}$ channels (TRP); however, the role of ion channels other than ORAI1 and STIM1 is controversial among researchers (*Feske, Wulff & Skolnik, 2015*; *Kotturi, Hunt & Jefferies, 2006*; *Nohara et al., 2015*; *Stokes, MacKenzie*

 

*& Sluyter, 2016*). Functional impairment of ORAI1 or STIM1 deficiency causes severe combined immunodeficiency (SCID) in mouse models and human patients, and various functional problems have been reported (*Prakriya & Lewis, 2015*; *Feske, Wulff & Skolnik, 2015*; *Nohara et al., 2015*).

Therefore, $I_{ORAI1}$ was measured using HEK293T cells overexpressing hORAI1 and hSTIM1 to observe the pharmacological effects of $\alpha$-mangostin. $\alpha$-Mangostin suppressed most ORAI1 currents at a concentration of 10 $\mu$M, and demonstrated extremely high potency with an IC$_{50}$ of $1.27 \pm 1.144$ $\mu$M (Fig. 1).

### Effects of $\alpha$-mangostin on K$^+$ channels (K$_V$1.3 and K$_{Ca}$3.1)

When Ca$^{2+}$, which is a divalent cation, continuously flows into the cell, the cell membrane voltage is depolarized, and the driving force to introduce calcium is weakened. During this time, K$_V$1.3 and K$_{Ca}$3.1 maintain the driving force for calcium influx by maintaining a negative cell membrane potential in T cells (*Cahalan & Chandy, 2009*). They are known to be regulated by immunological synapses after TCR activation (*Feske, Wulff & Skolnik, 2015*). Inhibition of K$_V$1.3 and K$_{Ca}$3.1 has been reported to attenuate calcium, and consequently, are involved in the activity and proliferation of T cells as well as the production and proliferation of cytokines (*Lam & Wulff, 2011*; *Koch Hansen et al., 2014*; *Feske, Skolnik & Prakriya, 2012*). K$_V$1.3 and K$_{Ca}$3.1 are typically mentioned as ion channels that regulate calcium influx in T cells, and research is being conducted to develop new immunosuppressive agents using them (*Lam & Wulff, 2011*; *Chandy & Norton, 2017*). In addition to the inhibition of ORAI1, whether $\alpha$-mangostin can inhibit K$_V$1.3 and K$_{Ca}$3.1, which controls calcium influx, were observed. As a result, 3 $\mu$M $\alpha$-mangostin, that could inhibit 70% ORAI1 by approximately 70%, inhibited K$_V$1.3 by $41.38 \pm 6.191$%, and K$_{Ca}$3.1 by $51.16 \pm 5.385$%. Therefore, the inhibitory effect of $\alpha$-mangostin on calcium influx appears to contribute to the inhibition of ORAI, as well as the inhibition of the potassium channels that regulate it.

### Inhibitory effect of $\alpha$-mangostin on calcium signaling

Intracellular calcium signaling depends on the inflow of calcium through ORAI1, and sufficient calcium inflow is achieved through hyperpolarization of the cell membrane by the activity of the K$^+$ ion channel (*Feske, Wulff & Skolnik, 2015*; *Cahalan & Chandy, 2009*; *Panyi, 2005*). In Jurkat T cells, 3 $\mu$M $\alpha$-mangostin inhibited thapsigargin-induced store-operated Ca$^{2+}$ entry (SOCE) by $77.14 \pm 6.600$% (Fig. 2) and ORAI1 expression by $70.51 \pm 6.185$%. $\alpha$-mangostin at this concentration showed a higher inhibition of calcium signaling, compared to inhibition of only ORAI1. This suggested that $\alpha$-mangostin suppressed the calcium signaling more effectively through inhibition of both the K$_V$1.3 and K$_{Ca}$3.1 ion channels and ORAI1. Intracellular calcium signaling regulates the degranulation in mast cells. Calcium signaling by antigens in RBL-2H3 cells is inhibited by $\alpha$-mangostin, possibly due to the inhibitory effect of $\alpha$-mangostin on K$_V$1.3 and K$_{Ca}$3.1, and ORAI1 (*Itoh et al., 2008*).

## Inhibitory effects of $\alpha$-mangostin on cytokine production

Anti-CD3/anti-CD28 stimulation generates calcium signals and activates NFAT to promote T cell proliferation and production of IL-2. Thus, we investigated whether $\alpha$-mangostin, with its inhibitory effects on ORAI1 and calcium signaling, could inhibit IL-2 production in Jurkat T cells. Prior to confirming the inhibition of IL-2 secretion, cytotoxicity of $\alpha$-mangostin was first evaluated, and most of the cells died at a concentration of 10 $\mu$M. Therefore, we examined the inhibitory effects of $\alpha$-mangostin on IL-2 production using concentrations up to 3 $\mu$M at which cells survived. At a concentration of 3 $\mu$M $\alpha$-mangostin inhibited IL-2 secretion of Jurkat T cells stimulated with anti-CD3 and anti-CD28 by 80.14 $\pm$ 3.987% at a concentration of 3 $\mu$M. The results obtained earlier confirmed that 3 $\mu$M $\alpha$-mangostin inhibited ORAI1 by approximately 70% and SOCE calcium signaling by approximately 77%, which is similar to the inhibition of IL-2 production observed with the same concentration of $\alpha$-mangostin. Moreover, according to the results of a previous study on the inhibitory effects of $\alpha$-mangostin on the secretion of cytokines in human peripheral blood mononuclear cells, $\alpha$-mangostin was reported to inhibit IL-2 secretion by concanavalin A (ConA) stimulation. ConA is known to play a role in producing IL-2 by stimulating TCRs, leading to $Ca^{2+}$ signaling and NFAT activation (*Fujita et al., 2019*; *Kasemwattanaroj et al., 2013*). This evidence indirectly supports the inhibitory effect on calcium signaling identified in this study.

## CONCLUSIONS

Most of the existing studies of $\alpha$-mangostin have been molecular investigations, focusing on the downstream signaling mechanisms. Only a few studies related to the underlying mechanisms such as ion channels and calcium signal regulation exist. We investigated whether $\alpha$-mangostin could regulate the activity of ion channels related to calcium signaling, and the underlying mechanism. We confirmed that $\alpha$-mangostin inhibited the influx of calcium by inhibiting ORAI1 and $K_V1.3$ and $K_{Ca}3.1$. In addition, it suppressed the secretion of IL-2, in Jurkat T cells. Therefore, this study revealed the effects of inhibiting the activity of immune cells by inhibiting the calcium ion channels, which play an important role in T cell activity.

### Funding

This research was supported by the Basic Science Research Program through the National Research Foundation of Korea (NRF) funded by the Ministry of Education (No. 2019R1I1A1A01059077) and also research was supported by the Convergence of Conventional Medicine and Traditional Korean Medicine R&D program funded by the Ministry of Health & Welfare (Korea) through the Korean Health Industry Development Institute (KHIDI) [grant number HI16C0766]. The funders had no role in study design, data collection and analysis, decision to publish, or preparation of the manuscript.

## Grant Disclosures

The following grant information was disclosed by the authors:

Basic Science Research Program through the National Research Foundation of Korea (NRF).

Ministry of Education: 2019R1I1A1A01059077.

Convergence of Conventional Medicine and Traditional Korean Medicine R&D.

Ministry of Health & Welfare (Korea) through the Korean Health Industry Development Institute (KHIDI): HI16C0766.

## Competing Interests

The authors declare there are no competing interests.

## Author Contributions

- Hyun Jong Kim and Seorin Park performed the experiments, prepared figures and/or tables, and approved the final draft.
- Hui Young Shin performed the experiments, authored or reviewed drafts of the paper, and approved the final draft.
- Yu Ran Nam and Phan Thi Lam Hong analyzed the data, prepared figures and/or tables, and approved the final draft.
- Young-Won Chin, Joo Hyun Nam and Woo Kyung Kim conceived and designed the experiments, authored or reviewed drafts of the paper, and approved the final draft.

## Data Availability

Raw data is available as Supplemental File.

## Supplemental Information

Supplemental information for this article can be found online at http://dx.doi.org/10.7717/peerj.10973#supplemental-information.

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
