# Peer review of "Inhibitory effects of α-Mangostin on T cell cytokine secretion via ORAI1 calcium channel and K+ channels inhibition"

_PeerJ, doi:10.7717/peerj.10973_

## Round 0.1 · original submission · Minor Revisions

Please address issues pointed by the reviewers and revise your manuscript accordingly.

·

Basic reporting

In this manuscript, Hyan Jong Kim et al., investigated the effects of  - Mangostin on ORAI1 calcium channels and potassium channels of immune cells. Using electrophysiology, they find that  - Mangostin inhibits ORAI1 calcium channels at IC50 of sub-micromolar concentration using both HEK293T and Jurkat T cell lines. They also find that  - Mangostin inhibits both voltage gated potassium channels Kv 1.3, and calcium-activated conductance channels KCa 3.1 that are essential for regulating membrane potential and for calcium signaling homeostasis. The authors looked at downstream effects of inhibition of calcium signaling on cytokine secretion and they find that IL-2 cytokine secretion was inhibited in Jurkat T cells. Thus, the authors concluded that inhibiting calcium channels and potassium channels is an underlying mechanism for anti-inflammatory effects of  - Mangostin. This is altogether a nice paper as authors provide direct evidence of  - Mangostin effects on ion channels in immune cells for the first-time using electrophysiology.

1. Background part of the paper requires rewriting.



2. I have annotated few sentences on a PDF file that needs to be rephrased or rewritten.

Experimental design

1.Please elaborate on Fura-2 Ca2+ imaging method section especially on how you use 340/380 ratio to calculate calcium concentration.

Validity of the findings

1. In Figure 1A, authors should provide a control trace for the entire length of the experiment. Also, in their Figure 1 legend, concentrations of  - Mangostin are in millimolar concentration, is it supposed to be milli molar or micro molar?

2. Cell viablity of Jurkat T cells went down by 90% at 10 M concentration (Figure 5). The authors look at ORAI1 channel inhibition at different concentration of  - Mangostin from 0.1-10 M, what about cell viability of HEK293T cells?

3.In Figure 4, if HEK293T cells are viable till 10 uM concentration, they should include I-V plot for other concentrations as in Figure 1 and determine IC50.

Reviewer 2 ·

Basic reporting

The study by Kim et al – Inhibitory effects of α-Mangostin on T cell cytokine secretion via ORAI1 calcium channel and K+ channels inhibition. The authors analyzed the inhibitory effect of α-mangostin on immune cell activity via inhibition of calcium and potassium ion channels expressed in immune cells and demonstrated the anti- inflammatory effect of α-mangostin through the regulation of calcium signals.

Experimental design

The objective of the study is reasonable. The methodology is well described, and experiments are performed with appropriate controls.

Validity of the findings

The data looks significant and conclusions are well drawn with statistical significance.

Additional comments

The authors use Jurkat T cells activated with anti CD3/CD28 antibodies and monitor the IL2 release following treatment with α-Mangostin. It would have been more convincing if authors would have monitored the primary T cell activation using CD3+T cells isolated from PBMCs and treated with α-Mangostin. It would be interesting, if authors can include in the discussion section, the speculative mechanism of α-Mangostin in modulating calcium signals in T cells.

---

## Round 0.2 · accepted · Accept

All remaining issues were adequately addressed and I am glad to accept your manuscript now.

·

Basic reporting

I appreciate Hyan Jong Kim et al., for addressing questions I raised. I also thank them for considering my suggestions and making necessary changes to the revised manuscript. it is well written now.

Experimental design

The methods described with sufficient detail.

Validity of the findings

Overall, it is a nice work on providing direct evidence of -Mangostin effects on ion channels in immune cells with carefully designed experiments.

Reviewer 2 ·

Basic reporting

The study by Kim et al – Inhibitory effects of α-Mangostin on T cell cytokine secretion via ORAI1 calcium channel and K+ channels inhibition. The authors analyzed the inhibitory effect of α-mangostin on immune cell activity via inhibition of calcium and potassium ion channels expressed in immune cells and demonstrated the anti- inflammatory effect of α-mangostin through the regulation of calcium signals.

Experimental design

Well described.

Validity of the findings

The authors provided the answers to the comments.